# Exploring Resilience and Its Determinants in the Forced Migration of Ukrainian Citizens: A Psychological Perspective

**DOI:** 10.3390/ijerph21111409

**Published:** 2024-10-24

**Authors:** Yaryna Andrushko, Stephanie T. Lanza

**Affiliations:** 1Edna Bennett Pierce Prevention Research Center, The Pennsylvania State University, University Park, PA 16802, USA; 2Department of Biobehavioral Health, The Pennsylvania State University, University Park, PA 16802, USA; slanza@psu.edu

**Keywords:** resilience, forced migration, trauma, acculturation, Ukrainian refugees

## Abstract

This study enhances the understanding of resilience in forced migration through a psychological lens, highlighting the importance of identifying resilience determinants and evidence-based interventions. By fostering resilience, policymakers and practitioners can support the well-being and adaptive capacities of forcibly displaced Ukrainians, promoting psychological recovery, social integration, and positive long-term outcomes for affected individuals and communities. To determine the key resilience indicators, survey data were collected in 2023 from *n* = 502 Ukrainian refugees living in the U.S. (M age = 27 years). Individuals reported various psychological factors and cultural experiences, revealing high resilience and low-stress tolerance among forced Ukrainian migrants in the U.S., along with a strong correlation between their adopted acculturation strategies and their resilience and levels of traumatization.

## 1. Introduction

The Russian invasion of Ukraine on 24 February 2022 changed the lives of Ukrainians forever. To date, nearly 300,000 Ukrainians have migrated to the United States, more than 271,000 of whom arrived under the Uniting for Ukraine program, which was developed in the United States for Ukrainians during the war [1,2]. Approximately 25,000 Ukrainians entered the United States at the beginning of the invasion, crossing the American land border from Mexico [3]. When a Ukrainian family comes to the United States, they undergo a process in which they become acclimated to the new culture and begin acculturation. During this process, they face various changes and losses, and they experience different types of emotions [4].

Psychological trauma is a worldwide phenomenon [5]. Healing from trauma does not always happen automatically when a traumatizing event ends and life returns to relative normality. Instead, there is often a remembrance, mourning, and healing period. Forced migration is one source of trauma. Migration is understood as a territorial movement of a population, associated with a change in place of residence [6]. In this context, we study the sociopsychological features of migrants, focusing on the challenges of adaptation, acculturation, and mental health.

### 1.1. Exploring Berry’s Framework and Its Application in Migrant Adjustment

Over 100 different theories of acculturation have been developed by academic researchers across various fields: for instance, the Acculturation Stress Model [7]; the Interactive Acculturation Model [8]; The Ecological Stress and Coping Framework [9]; and LaFromboise’s Bicultural Competence Model [10]. John W. Berry’s [11] framework is the most widely recognized model for understanding acculturation strategies. He identifies four primary approaches individuals or groups may adopt when navigating cultural adaptation: integration, which involves maintaining one’s original cultural identity while actively participating in the larger society; assimilation, where individuals embrace the host culture and relinquish their original cultural identity; separation, which occurs when individuals or groups choose to retain their original culture while minimizing interaction with the larger society; and marginalization, a strategy in which individuals neither preserve their original culture nor engage with the new culture. Among these, Berry’s model of acculturation [12] has been extensively utilized to comprehend the acculturation experiences of immigrants and minority groups. The model categorizes individual adjustment strategies using two dimensions. The first pertains to whether individuals choose to preserve or adapt their cultural identity. This involves assessing the value of preserving one’s cultural identity and characteristics. The second dimension pertains to the degree of involvement or disengagement with the host culture and whether there is value in maintaining relationships with the new, larger society [12]. Acculturation refers to “the processes by which groups or individuals adjust the social and cultural values, ideas, beliefs, and behavioral patterns of their culture of origin to those of a different culture” [11].

### 1.2. Trauma and Resilience in Forced Migration

We give a synthesized explanation of trauma that draws from the general psychological and clinical literature, reflecting common themes in trauma research. Trauma is a psychological and emotional response to an event or experience that is deeply distressing or disturbing, often involving a threat to life or safety. It can result from a single incident, such as an accident or attack, or from ongoing stressors, such as abuse, war, or forced migration. Trauma can have long-lasting effects on an individual’s mental, emotional, and physical well-being, leading to symptoms such as anxiety, depression, intrusive thoughts, hyperarousal, or emotional numbness. In its broader sense, trauma encompasses both the event and the personal impact it has on an individual’s psychological and physiological functioning.

Understanding trauma as a result of forced migration in connection with war—and the formation of individual resilience in this context—requires clear conceptual foundations. Trauma from forced migration must be studied from a multidisciplinary perspective, including fields such as psychology, sociology [13], sociodemography [14], cultural anthropology [15], and communication [16]. Several clinical perspectives can also be brought to bear on the study of general migration and forced migration [17,18,19]. Trauma is not limited to violent or life-threatening events; it can also arise from the psychological burden of adjusting to a new environment and struggling with communication barriers, which may lead to chronic stress. Migrants may feel isolated or unable to access essential services, healthcare, or education. The inability to communicate effectively can exacerbate feelings of helplessness, confusion, and marginalization [20,21]. Understanding the type and context of trauma is essential to contextualize the resilience mechanisms that migrants adopt [22].

Theoretical approaches to trauma highlight both positive and negative dimensions of traumatic experiences [23]. The negative aspects, such as loss, vulnerability, and psychological or physical harm, are well-documented. Trauma can inflict lasting damage on an individual’s mental and physical health, leading to long-term suffering. However, trauma also has positive dimensions. For example, following physical or psychological injury, individuals may experience a shift in their perception of life’s value. What was once taken for granted can gain significance, and a new appreciation for ordinary life may emerge. This phenomenon suggests that, while trauma disrupts life, it can also foster resilience and personal growth. To contextualize these ideas within trauma theory Paulson [24] identifies six broad theoretical frameworks for understanding trauma in civilians affected by war: psychiatric theory, developmental approaches, psychoanalysis, family-oriented approaches, learning theory, and cognitive behavioral therapy. While all these frameworks contribute valuable insights, our study primarily draws on dialectical behavioral therapy (DBT), due to its focus on restructuring maladaptive thought patterns that emerge after trauma. This is particularly relevant to the present research, as many Ukrainian refugees face ongoing cognitive and emotional challenges due to the sudden and violent nature of their displacement.

In this context, Herman’s [25] model of post-traumatic stress disorder (PTSD), which organizes symptoms into three categories—hyperarousal, intrusion, and constriction—is particularly relevant. Hyperarousal refers to a persistent state of heightened alertness and expectation of danger, intrusion describes the involuntary and recurrent re-experiencing of traumatic events, and constriction is characterized by emotional numbness or a sense of helplessness. These categories are critical to understanding refugees’ psychological profiles. By grounding our research in Herman’s framework, we aim to explore how these PTSD symptoms manifest in the Ukrainian refugee population and how psychological interventions can mitigate these effects.

It is obvious that forced migration is associated with a sense of loss of the land where one was born and raised and with certainty about one’s future. This is coupled with uncertainty about whether that future involves returning to one’s home. As a result of the forced abandonment of the home, the process of developing the human self in the context of one’s native land and culture is interrupted. The losses of loved ones, work, social status, and heirlooms are all salient. Therefore, after satisfying basic needs (food, safety, housing), migrants require psychological support to help mitigate feelings of isolation and loneliness, build new connections, and integrate into a new society with what is often a changed identity. For example, during forced migration, a person may feel survivor guilt—feelings of guilt and shame simply for surviving when others did not [26,27]. This can penetrate deeply into the consciousness and have an impact not only on the individual but also on future generations [28].

Ukrainian refugees likely experienced acute trauma from myriad experiences, such as witnessing violent conflict, experiencing displacement, the process of fleeing one’s home, separation from loved ones, and the abrupt loss of safety and stability. In addition, some refugees also have experienced more chronic, complex trauma, especially those who had pre-existing exposure to regional instability, political tensions, or economic hardships before the war [29]. Those who lived through the annexation of Crimea in 2014 or faced longstanding economic and social challenges may be affected by layered trauma that predates the current war [30]. Finally, post-migration stressors such as family separation, difficulty adjusting to a new culture, language barriers, and economic instability can extend or exacerbate the original trauma [27].

Resilience often emerges as a response to trauma, and understanding the nature of the trauma faced by migrants (acute or complex) can provide insight into how individuals rebuild their lives. According to the American Psychological Association [31], resilience is both the process and the outcome of successfully adapting to challenging life experiences. Resilience theory directs attention toward identifying positive contextual, social, and individual factors that can mitigate or disrupt developmental pathways, leading to problematic behaviors, psychological distress, and negative health outcomes [32,33,34,35,36]. The term resilience has historically been used to describe various behaviors, circumstances, and achievements [37] that enable one to survive trying circumstances. In addition, early interpretations of resilience theory associate it with personal traits such as humor and intelligence. The current consensus is that resilience is an innate response to stress that allows people to respond positively and function effectively in their environment [38,39,40]. Many studies have established the role and importance of resilience for the formation of a healthy personality [17,41,42,43,44] and the ability to recover from trauma and challenges, such as those that arise because of forced migration [45]. Research is needed to develop and test strategies to build resilience, mitigate damage from trauma, and promote individuals’ potential. Oviedo et al. [46] points out that refugees’ resilience greatly improves when they encounter welcoming people, when they can maintain communication with loved ones in their home country, and when they can develop a rich inner life, often in the form of traditional prayer.

Independent analyses by Lazos [47], Masten [48], and Richardson [49] concur on which factors lead to the formation of resilience: (1) experience of a traumatic event, followed by the activation of (2) protective factors and (3) vulnerability factors, and then the (4) interaction between protective and vulnerability factors to mitigate the negative effects of the traumatic event. Simultaneously, Hamby, Grych, and Banyard [50] note that personality resilience occurs in three steps: traumatic and/or stressful event, healthy functioning after a traumatic situation, and mechanisms that allow one to resist stress and recover from trauma. In this view, resilience is based on three factors: risk, protection, and vulnerability. Charney and Nemeroff [51] and, later, Hellerstein [52] describe components of resilience: physical; psychological; and activating social networks, including confiding relationships and adequate external supports; challenging oneself; looking for meaning through involvement; and learning. Additionally, daily stressors were found to mediate this. The research also highlighted significant differences in anxiety levels between people with and without children, specifically in the overall sample and the Ukrainian sample, but not among Romanian civilians [53].

### 1.3. Functional, Health, and Spiritual Adaptation Among Ukrainian Refugees

The process of leaving one’s country as a result of war entails several processes. One must adapt to new socio-cultural conditions; adjust to physiological problems (e.g., disruption of sleep, eating disorders); deal with loss of meaning and value in life, which can be experienced as an existential vacuum; and handle frustration, apathy, dissatisfaction, or internal conflict, which can lead to neurosis. Adapting to these conditions can lead to resilience. Below, we propose three facets of resilience that displaced people may work toward as they move toward healing. Individuals may develop these facets sequentially or simultaneously.

Facet 1. To achieve functional resilience, individuals must adapt to new sociocultural conditions and environmental features. Most people who are forced to leave their homes are faced with the need to create new, comfortable living conditions including a house, furniture, utilities, employment, and transportation.

Facet 2. Individuals may pursue health resilience, which encompasses good psychological and physical health in the face of significant displacement-induced stress [54,55,56].

Facet 3. Spiritual resilience—the feelings of a meaningful existence, understanding, and feeling the value of life—is also critical. Working toward a goal can add meaning to one’s life, provide motivation to move forward, and act as a support [57]. Frankl’s insight [57], drawn from his own experience surviving Nazi concentration camps, emphasizes the importance of finding meaning and faith in the context and content of one’s life, even amidst extreme suffering. While this idea may not emerge from a formal psychological theory, it is rooted in Frankl’s [58] personal journey and philosophy, which holds that meaning can be a source of resilience in the face of trauma.

The literature on trauma and forced migration reveals the complex psychological challenges that arise from displacement, including acute and complex trauma, ongoing stressors, and the loss of identity and security. While trauma can lead to psychological distress, it can also foster resilience. In the case of migrants, resilience is not only a psychological outcome, but also a process influenced by health, social support, cultural integration, and spiritual practices. Drawing on these insights, our research seeks to explore how these dynamics manifest in Ukrainian refugees in the United States. Specifically, we aim to understand how functional, healthy, and spiritual resilience emerge in response to forced migration and how they interact to support psychological well-being in a new cultural context.

### 1.4. Objectives and Contributions

This study aims to explore the psychological conditions of Ukrainian refugees in the U.S., with a focus on their trauma responses and resilience-building processes. Using Berry’s model of acculturation [11] and Herman’s model of PTSD [25], we seek to understand how refugees navigate the intersection of cultural adaptation, trauma, and resilience. Specifically, we examine three facets of resilience—functional, health, and spiritual resilience—and propose strategies for fostering resilience within this population. The contribution of this study lies in its integration of trauma theory, acculturation models, and resilience frameworks to provide a comprehensive analysis of Ukrainian refugees’ psychological adaptation in the context of forced migration.

## 2. Materials and Methods

### 2.1. Participants and Procedures

Recruitment of participants took place by direct e-mail, via flyers posted digitally on Ukrainian Facebook groups (e.g., Ukrainians in U.S.), and through flyers posted in public places, such as a library and a coffee shop. Recruitment occurred from April to June 2023. Advertisements included a link that took individuals to a screening questionnaire. To be eligible to participate in the study, individuals who were forced to migrate from Ukraine must have resided in the United States for no more than 10 years, not possess a Green Card, not be married to an American citizen, and be age 18 years or older. Eligible individuals provided written informed consent prior to completing the survey, then received a $10 gift card as compensation. Among those who completed the screening questionnaire, 88% were eligible; among these individuals, 85% completed the survey, resulting in *n* = 502 individuals in the study. The study was conducted online using RedCap (The Pennsylvania State University, State College, USA) in native Ukrainian language. Data collection took place during May 2023 and June 2023.

### 2.2. Measures

Participants provided data on demographic information, including their age, sex, living arrangement, marital status, level of education, religion, profession (field of activity), language preferences, level of English, time living in the United States, relatives in the United States planning to return to Ukraine, and friends and acquaintances in the United States. To measure functional resilience, we asked participants about their living conditions, accommodation, and environment.

Psychological components of resilience were assessed to determine the foundational experience of forced migration. The 25-item Resilience Scale [59] was used to assess participants’ resilience. This scale used questions like “I can be on my own if I have to” and “I do not dwell on things that I can’t do anything about.” Items on the resilience scale were summed to form an overall resilience score, with scores below 65 indicating low resilience, scores between 65 and 81 indicating moderate resilience, and scores above 81 indicating high resilience. The Connor–Davidson Resilience Scale-10 (CD-RISC-10) [60] was used to measure stress resistance. This 10-item scale included questions such as, “I can adapt to changes” and “I can overcome any obstacles in my path.” Total scores were calculated by summing all 10 items, with a higher score indicating higher resilience. Stress resilience refers to an individual’s capacity to adapt to stressors and maintain psychological well-being in the face of adversity. Stress resilience can be considered a subset of resilience. General resilience encompasses a wider range of responses to various types of adversity, including physical, emotional, and social challenges. In contrast, stress resilience focuses on an individual’s responses to stress and stress-related situations. The two constructs can diverge in their measurement and implications; for instance, items from the resilience scale, such as “I do not dwell on things that I can’t do anything about,” emphasize acceptance and emotional processing, whereas stress resilience items like “I can overcome any obstacles in my path” highlight proactive coping and problem-solving. This distinction suggests that while both constructs contribute to an individual’s ability to cope with adversity, they do so in different ways [61].

Spiritual components of resilience, including five dimensions—finding the meaning of life, religiosity, identity, value of life, and acculturation—were assessed using the Mutual Intercultural Relations in Plural Societies (MIRIPS) Questionnaire [62]. Some measures are available in two formats, one for use with non-dominant samples (e.g., immigrant and ethnocultural groups, national/regional minorities) and the other for use with the dominant national (or dominant regional) samples; we used the version for non-dominant samples to assess acculturation. The Mutual Intercultural Relations in Plural Societies (MIRIPS) Questionnaire includes several subscales that assess key dimensions relevant to individuals affected by forced migration. Here’s a concise overview:-Religion and Level of Religiosity: Assesses the importance of religious beliefs and practices in coping with challenges and fostering resilience.-Self-Satisfaction: Measures individuals’ perceptions of their self-worth and fulfillment after displacement.-Life Satisfaction: Evaluates overall satisfaction with life, considering various domains like relationships and health.-Attitude Toward Identity: Explores feelings regarding personal and cultural identity, particularly in the context of migration.-Perceived Discrimination: Assesses experiences of unfair treatment based on cultural background, which can impact mental health.-Intergroup Relations: Evaluates the quality of interactions with individuals from different cultural backgrounds, highlighting social integration.-Cultural Identity: Measures the significance of individuals’ ties to their original culture while adapting to a new environment.-Level of Depression: Assesses the presence and severity of depressive symptoms related to trauma and displacement.-Level of Anxiety: Evaluates the intensity of anxiety symptoms, which may arise from the uncertainties of migration.

These subscales provide a comprehensive view of the challenges faced by displaced individuals, particularly Ukrainian refugees. They reflect how religion, identity, and social dynamics influence resilience and psychological well-being, while also addressing mental health issues like depression and anxiety. The interplay among these dimensions offers valuable insights into the acculturation process and the support mechanisms that can aid in recovery and adaptation.

In the four-dimensional approach, we directly measure each of Berry’s four acculturation strategies. This involves creating items that correspond to the specific strategies: assimilation, separation, integration, and marginalization. Each strategy is measured by several questions on a Likert scale (e.g., 1 = “strongly disagree” to 5 = “strongly agree”). For each dimension (assimilation, integration, separation, marginalization), we calculated a score by averaging the responses for items related to that dimension. Integration was measured by items related to engagement with both the host and heritage culture (e.g., “I want to maintain my cultural traditions, but also adopt aspects of the new culture.”). Assimilation was measured by items related to adopting the host culture and letting go of the heritage culture (e.g., “I prefer to adopt the local culture and let go of my original cultural practices.”). Separation was measured by items related to maintaining the heritage culture and avoiding the host culture (e.g., “I prefer to interact only with people from my original culture.”). Finally, marginalization was measured by items related to disengagement from both cultures (e.g., “I don’t feel connected to either my original culture or the new one.”).

Additional data were collected to assess the effects of the traumatic experience of forced migration. The Posttraumatic Stress Disorder Checklist for DSM-5 (PCL-5) [63] was used to assess symptoms of post-traumatic stress disorder. This included Criterion A, which is a description of the traumatic event, and a 20-item scale comprising four subscales: Items 1–5 assessed Criterion B, intrusion symptoms; Items 6–7 assessed Criterion C, avoidance symptoms; Items 8–14 assessed Criterion D, negative thoughts and emotions; and Items 15–20 assessed Criterion E, symptoms of excessive reactivity. The impact of specific events or experiences on an individual’s psychological well-being was assessed using the 22-item Impact of Events Scale-Revised (IES-R) [64]. The scale included three subscales to assess avoidance, intrusion, and hyperarousal. The items on the IES are worded to refer to a specific traumatic event, allowing participants to reflect on their personal experiences. This ensures that the scale measures trauma-related symptoms in direct relation to the most salient events experienced by the individual, in this case, forced migration and war-related experiences.

### 2.3. Data Analysis

For each scale, reliability was calculated, and scale scores were created; we then obtained descriptive statistics for all variables used in the current analysis. Linear regression analysis was used to estimate a set of models predicting resilience as a function of nine predictors. Predictors were included in separate models to quantify the relative amount of variance in resilience explained by each. Predictors included cultural safety, sociocultural maladaptation, stress resistance, depression, perceived physical safety, anxiety, integration, and attitude toward identity. All analyses were conducted using SPSS 30 software.

The questionnaire consisted of 190 questions and took, on average, approximately 30 min to complete. However, participants were informed that they could stop at any point, allowing them the flexibility to finish when they felt comfortable. This flexibility was important to mitigate potential response fatigue, ensuring that participants only continued as long as they were willing and able.

## 3. Results

### 3.1. Characteristics of Participants

First, we summarized the socio-demographic characteristics of the sample from the demographic survey. Descriptive statistics for the 502 participants (46% male, 54% female; *M* age 27 years, range 18–58 years) are shown in Table 1. Among these individuals, 4.6% have lived in the United States for less than one year, 66% for one to two years, 4.6% for four to six years, 2.2% for six to nine years, and 0.2% for more than nine years. A significant proportion of individuals reported wanting to return to Ukraine (35%); the same proportion reported wanting to stay in their current location, whereas the remaining individuals (30%) had not yet decided. A small proportion of individuals indicated having relatives living in the United States (13%) or living with their relatives in the United States (2%). The most common living arrangement was living alone (47%), with other arrangements including living with American acquaintances/friends (28.0%), with Ukrainian acquaintances/friends (5.8%), with a partner (28.0%), or another option (2.5%).

In terms of education, about one-fifth of respondents had completed some college (24%); 23% had completed secondary vocational education; 20% had completed higher education (bachelor’s or master’s degree); 15% had completed some secondary vocational education; 14% had finished high school; 2% reported having a Ph.D. degree; and 2% had attended high school but did not finish.

Individuals reported the following current employment: office worker (e.g., clerk, manager, secretary; 44%), skilled worker (e.g., technician, carpenter, hairdresser, tailor; 34%), unskilled labor (e.g., cleaner, loader, laborer; 12%), professional specialist (e.g., doctor, lawyer, teacher, manager, administrator; 5%), or not currently working (5%). Those not working reported various arrangements including being currently unemployed, a homemaker, or a student.

In terms of marital status, participants were most likely to report being married (43%) or single (38%), with a smaller proportion reporting being divorced (8%) or having a partner (10%). Among the unmarried participants, 62% expressed a desire to build relationships with individuals of their own nationality, 26% stated that the nationality of their partner is not important to them, and only 12% of the respondents expressed a lack of desire to form relationships with Ukrainians.

The religious affiliations reported by participants were Christianity (61%), Islam (15%), Judaism (9%), Buddhism (2%), Atheism (12%), or Other (e.g., “I believe in the universe,” “I am upset with God”; 0.4%). In terms of their level of religiosity, 13% reported “I am a convinced atheist,” 9% reported “I am indifferent to religion,” 14% reported “I believe in the existence of higher powers,” 35% reported “I am a religious person,” and 29% reported “I am a religious person and try to observe all the rituals of my religion.”

### 3.2. Emotions and Thoughts About War and Trauma

Regarding their thoughts on the current war in Ukraine, most (61%) of the respondents reported avoiding thoughts, emotions, and feelings about the war and the situation; 46% reported struggling with thoughts of guilt and self-blame for not being in Ukraine; 46% did not have any thought of guilt; and 8% did not know what they were feeling. A large majority (76%) of respondents reported their experience of forced migration as traumatic, while a minority (20%) did not consider their experience to be traumatic; 4% were unsure. When asked to indicate the most traumatic aspects of migration, 40% indicated the language, 36% indicated the new culture, 50% indicated both the language and the new culture, 47% indicated the significant distance from home and family, 22% indicated all the above; and 1% indicated another aspect as most traumatic, such as feeling safer at home, access to healthcare, starting life over from scratch, vulnerability of a certain social stratum, and loss of their home and everything in it.

In terms of languages, 4% speak English at home, 53% speak Ukrainian at home, and 44% report being bilingual and communicating in both English and Ukrainian. The participants had varying levels of English language proficiency: don’t speak (5%), basic level (36%), intermediate (32%), good (22%), fluent (5%). Similarly, they had varying levels of English writing proficiency: can’t write (23%), basic level (24%), intermediate (24%), good (14%), fluent (15%). In terms of their English reading proficiency, they reported the following levels: can’t read (19%), basic level (21%), intermediate (29%), good (20%), fluent (11%). English comprehension had a similar distribution: don’t understand (10%), basic level (35%), intermediate (23%), good (23%), fluent (10%).

Socialization and the ability to communicate in a new society are fundamental. When asked to describe their social network in the United States, responses varied: almost all people around me are of a different nationality (15%); most people around me are of a different nationality (39%); around me, there is an approximately equal number of people of my nationality and other nationalities (37%); most people around me are of my nationality (10%); and almost all people around me are of my nationality (2%). Regarding current friendships while living in the United States, participants were asked about the number of close American friends they had. Responses varied, with 6% reporting no American friends, 32% reporting having one close American friend, 46% reporting two to three close friends, and 15% reporting more than three. These findings highlight the diversity in participants’ social networks and their varying degrees of integration into American society. More participants reported close Ukrainian friends (9% reported 0, 23% reported only 1, 29% reported 2–3, and 39% reported more than 3). The participants also reported having close friends of other nationalities (5% reported 0, 18% reported 1, 49% reported 2–3, and 28% reported more than 3). The participants reported spending more time with their close Ukrainian friends than their close American friends. Specifically, they reported spending time with their American friends never (4%), rarely (21%), sometimes (42%), or often/every day (23%); they reported spending time with their Ukrainian friends never (8%), rarely (21%), sometimes (23%), or often/every day (49%).

### 3.3. Resilience, Trauma, and Acculturation Strategies

The participants reported a high mean level of resilience (*M* = 106) and a low mean level of stress resistance (*M* = 26). In terms of PTSD symptoms, higher scores were observed for the avoidance (*M* = 19) and intrusion (*M* = 20) scales, indicating common experiences of reliving traumatic events through nightmares, brief flashes of memory, reluctance to talk about the war or their migration, ongoing events, and personal experiences. The overall trauma index M = 47, which is higher than the average total score on this scale. Based on the PTSD Checklist, respondents exhibited the highest scores in Cluster D (*M* = 16), which describes negative thoughts and emotions related to the traumatic event. The participants also scored high on Cluster E (M = 14), characterized by excessive personality reactivity. The most common acculturation strategy was assimilation, where individuals associated themselves with the new culture and rejected the culture of the ethnic minority group to which they belonged.

### 3.4. Interrelated Elements of Resilience and Stress Resistance in the Context of Forced Migration

Correlations were calculated to explore the relationships of resilience and stress resistance with other covariates (see Table 2). Many of the correlation coefficient covariates with the resilience outcomes were statistically significant. One of our key findings here is that some individuals may experience cultural stress or identity struggles over time, especially if they face challenges in assimilating into American society. This prolonged stress could potentially erode resilience. There was an unexpected finding that the correlation between the level of religion and the resilience scale was negative (b = −1.96, *p* < 0.01), such that stronger religious beliefs were associated with lower resilience.

### 3.5. Correlates of Overall Resilience

Table 3 shows the unstandardized and standardized regression coefficients for individual covariates predicting scores on the resilience scale. Depressive and anxiety states are associated with lower levels of resilience. The more these conditions prevail in a person, the more difficult it is to maintain stress resistance and adapt to new conditions. This is a natural process because emotional exhaustion greatly complicates a person’s ability to deal with stress and overcome life’s difficulties.

Interestingly, we found a negative association between the acculturation strategy of integration and resilience (b = −0.137, *p* = 0.011): the more a person integrated into a new environment, the lower their level of resilience.

## 4. Discussion

Refugees often face significant challenges, including displacement, loss of home and community, and exposure to trauma. In the context of forced migration, a high level of resilience may correspond to a greater sense of safety and satisfaction with oneself and one’s life, as well as a lower level of anxiety and other mental health problems.

As the war in Ukraine continues, so does the traumatization of the Ukrainian people. Personal knowledge, skills, and traits are decisive factors in how one adapts to new living conditions; thus, resilience is critical for individuals facing the unique challenges of forced migration. In such circumstances, the presence of supportive individuals and communities can have a profound suggested positive effect on refugees’ well-being and ability to adapt.

In our study, higher levels of resilience were associated with praying, family support, and awareness about specific situations. On the other hand, individuals with lower levels of resilience may be more prone to anxiety, as they are less able to effectively manage stress and adapt to negative situations. Here is one possible interpretation. In terms of the development of depression, individuals with higher levels of resilience may be more resistant to developing depressive symptoms, as they have a greater ability to adapt to life adversities and recover more quickly from stress. Conversely, individuals with weaker resilience may be more vulnerable to depression, as they have fewer resources to overcome challenging situations. Our results are consistent with other research documenting high levels of depression and anxiety symptoms among Ukrainians who have been forced to migrate [65].

In addition, our findings suggest a relationship between a person’s resilience and the context of their place of residence, which corresponds to different living conditions. Correlation analyses indicated that individuals who lack a familiar support network, such as relatives or close friends, tend to exhibit lower levels of resilience compared to those with established social connections, regardless of their geographical distance from their home country. This finding aligns with Berry and Sam’s [66] framework of acculturation, which emphasizes the importance of social support in fostering resilience, as well as Kim’s [67] concept of host receptivity, highlighting the role of a welcoming environment in promoting adaptive coping strategies among migrants. Moreover, if integration occurs in a society where a person faces discrimination or exclusion, it can further deepen feelings of isolation and stress. As a result, the more effort a person puts into integration, the more they can feel the pressure, which leads to a decrease in resistance.

The adaptation and acculturalization process can be especially difficult for those who have experienced traumatic events related to the war. The high level of stress that accompanies attempts at integration can deplete a person’s internal resources, challenging their resilience and reducing their ability to navigate additional obstacles.

Additional significant predictors of resistance were cultural safety, physical safety, attitudes toward identity, and sociocultural maladaptation. Ensuring physical safety is a top priority to facilitate resilience, especially in the context of forced migration, where people face new and often unknown challenges. If a person feels culturally safe, has a positive attitude towards their identity, and successfully adapts to the new socio-cultural environment, they are likely to have a higher level of resilience. Conversely, the absence of these factors can make a person more vulnerable to stress and other negative life circumstances.

Individuals may develop the ability to cope with stress, adversity, and uncertainty, which can contribute to well-being and resilience [46,68,69,70]. War often forces individuals to confront the fragility of life and the impermanence of circumstances. This can lead to a profound shift in perspective, prompting individuals to re-evaluate their priorities, values, and goals. Overcoming the trauma of war can foster a profound appreciation for life and its intrinsic value. The original trauma of war, the challenges of cultural adjustment are a combination of that shapes Ukrainian resilience. This heightened awareness can influence their approach to relationships, opportunities, and personal fulfillment [71]. However, Helmreich et al. [72] write about support in identifying self-efficacy resources (e.g., social connections) and reminding oneself of previous achievements. Individual characteristics such as intelligence, feeling that life is meaningful, and manageability of life also may be important components in the development of resilience in our future research. Scientists [73] suggest that important facets of trauma recovery include developing an understanding of external life challenges, personal beliefs, and emotions, and being encouraged to identify and use personal resources (either internal or external). Thus, psychological resilience is explained in various degrees by three interpretations: as a person’s trait or ability to overcome stress, as a coping process, and as an adaptive and protective mechanism that helps resist stress and/or adapt after experiencing a traumatic event.

When a person is forced to leave his home because of the war, they enter a new environment where they try to integrate. This environment often has completely different cultural and social norms. Integration in such a situation can include learning a new language, finding a job, establishing social contacts, and adapting to new realities of life, which is often accompanied by great stress.

People who have experienced traumatic events related to war may have varying levels of resilience. Resilience can influence how a person responds to traumatic events and how quickly they can recover from them. Individuals with higher levels of resilience may be better equipped to adapt to stressful situations accompanying war trauma. People with low resilience may be more prone to developing PTSD and exhibiting more intense symptoms, such as intrusive memories of traumatic events, hyperarousal, feelings of detachment, and so on. Similarly, individuals with higher levels of resilience may display less intense PTSD symptoms or experience a faster path to recovery. On the other hand, individuals who do not have PTSD may be more able to be resilient. Indeed, individuals with higher levels of multifaceted resilience may be more able to adapt to new conditions and learn to communicate and interact effectively with a new culture. Resilience can enhance one’s ability to navigate cultural differences, cope with challenges, and build positive relationships in a new cultural context. People with higher levels of resilience may exhibit a greater ability to learn a new language, which can help mitigate stress corresponding to the cultural shift. Resilience may increase one’s persistence and motivation when trying to overcome the many obstacles faced by forced migrants who are navigating so many changes.

This study has several limitations. The sample may not be representative of the broader population of Ukrainian forced migrants due to potential biases in recruitment methods. For instance, participants were selected from specific communities; the findings might not capture the diversity of experiences among all Ukrainian forced migrants. While 502 participants provide a substantial dataset, it may still not encompass the full range of experiences and resilience factors within the larger population of Ukrainian forced migrants. This limitation can affect the generalizability of the findings. The unique cultural and contextual factors affecting Ukrainian forced migrants may limit the applicability of the findings to other populations. Factors influencing resilience in Ukrainian migrants might differ significantly from those affecting forced migrants from other cultural or national backgrounds. The timing of the study can influence its findings: psychological impacts and resilience factors may vary at different stages of the migration process. The experiences of those recently displaced might differ from those living in the host country for a longer period. The methods used to collect data, such as surveys and self-reported measures, can introduce biases like social desirability bias, where participants may present themselves more favorably. These biases can impact the reliability and validity of the findings. In the context of this study, the findings and implications specifically apply to individuals experiencing trauma due to forced migration in war situations, while also noting that these insights may be relevant to other high-trauma scenarios requiring resilience.

## 5. Conclusions

Our findings suggest that resilience is multifaceted, and thus a comprehensive, multifaceted approach is crucial when developing programs to help forced migrants build resilience. Supporting Ukrainian refugees, and forced migrants in general, will require sustained effort, time, and investment. The results of this study—while focused on Ukrainian refugees—can be broadly applied to migrants from various backgrounds, as the psychological impacts of forced migration often transcend cultural and geographical boundaries.

This research highlights the importance of addressing multiple dimensions of resilience—functional, psychological, and spiritual—in forced migrant populations. By integrating models of acculturation and trauma, we offer a framework for understanding the layered trauma experiences faced by Ukrainian refugees. Our contribution lies in demonstrating that a resilience-building program must address not only immediate practical needs (e.g., housing, language support) but also long-term psychological and emotional recovery strategies.

The study emphasizes that resilience is not a one-size-fits-all construct; instead, it encompasses various aspects of human functioning, including mental health, physical well-being, and spiritual fulfillment. Programs designed to aid forced migrants must therefore consider these diverse needs. Psychological interventions, such as dialectical behavioral therapy (DBT), can help individuals manage the cognitive and emotional challenges of displacement, while community-based support networks may foster a sense of belonging and reduce isolation. Additionally, fostering spiritual resilience through culturally sensitive practices—whether through religious support or other forms of spiritual expression—can provide vital psychological relief.

Future research should focus on identifying those individuals who are at heightened risk of enduring psychological trauma, enabling more targeted and tailored interventions. Given the variability in how individuals experience and cope with trauma, understanding personal and contextual factors—such as prior exposure to conflict, family structure, and social support—can help predict resilience outcomes. Research might also explore the long-term effects of acculturation strategies, such as integration or separation, on the mental health of forced migrants.

Moreover, as global forced migration is expected to rise due to factors like climate change and political instability, future studies should investigate how resilience strategies can be adapted for different groups facing various types of displacement. This includes not only war refugees but also those displaced due to natural disasters, economic crises, or persecution. Comparative studies between different refugee populations would help to generalize findings across broader migrant groups and refine intervention programs to cater to their specific needs.

Finally, while our study underscores the importance of individual resilience, we also stress the need for systemic and international efforts. Addressing the root causes of forced migration—whether conflict, economic deprivation, or climate change—requires global cooperation. Investments in conflict prevention, international partnerships, and sustainable development are essential to reducing forced migration at its source. In the long term, such measures can help ensure that individuals are not forced to leave their homes, but can live in peace, dignity, and prosperity in the communities of their choosing.

## Figures and Tables

**Table 1 ijerph-21-01409-t001:** Descriptive statistics of key study variables (*n* = 502).

Variables	Mean	Std	Min	Max	Range
Resilience Scale	105.90	18.46	66	161	25–175
Stress Resistance Scale	25.51	4.56	11	41	10–50
Criterion B (Symptoms of Intrusion)	11.74	2.91	0	21	0–24
Criterion C (Avoidance Symptoms)	4.74	1.62	0	10	0–12
Criterion D (Negative Thoughts and Emotions)	16.39	3.97	1	29	0–36
Criterion E (Excessive Reactivity)	14.02	3.75	1	25	0–30
Overall Trauma Rate	46.89	9.67	3	79	0–80
Avoiding	19.06	4.13	7	35	0–35
Intrusion	19.54	4.38	6	36	0–36
Hyperarousal	14.27	3.49	1	25	0–30
Attitude to Identity	35.46	6.53	16	54	12–60
Cultural Safety	16.25	2.97	3	23	0–24
Economic Security	12.86	2.54	2	18	0–18
Physical Security	12.81	2.55	3	18	0–18
Integral Indicator of Safety	41.93	6.36	17	56	0–60
Assimilation	9.23	2.50	2	16	0–16
Integration	6.92	2.23	2	12	0–12
Separation	4.61	1.48	1	10	0–10
Marginalization	7.43	1.78	3	12	0–12
Level of Depression	14.23	3.75	3	26	0–30
Level of Anxiety	14.00	3.58	2	26	0–30

**Table 2 ijerph-21-01409-t002:** Associations of resilience and stress resistance with individual characteristics.

Individual Characteristics	Resilience Scale	Stress Resistance Scale
Demographic and Cultural Variables		
Time of Residence in the United States	−0.135 **	0.051
Live with Relatives	0.344 **	−0.146
Language as a Trauma	−0.109 *	0.143 **
Culture and Language as a Traumatic Category	−0.102 *	0.067
Relatives in the United States	−0.091 *	0.160 **
Sociocultural Maladaptation	−0.376 **	0.154
Cultural Safety	0.454 **	0.130 **
Integration	−0.156 **	0.075
Indicators of Spiritual Resilience		
Attitude Toward Identity	0.181 **	0.226 **
Religion	−0.105 *	0.102 **
Level of Religiosity	−0.196 **	0.201 **
Self-Satisfaction	0.159 **	0.206 **
Life Satisfaction	−0.249 **	0.151 **
Indicators of Psychological Resilience (Mental Health and Well-Being)		
Stress Resistance	0.264 **	1.000
Level of Anxiety	−0.296 **	0.121 **
Level of Depression	−0.301 **	0.030
Perception of Discrimination	0.243	−0.200 **
Excessive Reactivity (Criterion E)	−0.203 **	0.0303
Negative Thoughts and Emotions (Criterion D)	−0.238 **	0.104 *
Avoidance Symptoms (Criterion C)	−0.186 **	0.087
Symptoms of Intrusion (Criterion B)	−0.217 **	0.051
Hyperarousal	−0.157 **	0.097 *
Trauma, Safety, and Security		
Overall Trauma Level	−0.273 **	0.151 **
Integral Indicator of Safety	0.405 **	−0.173 **
Physical Safety	0.236 **	0.010
Economic Security	0.245 **	0.061

** Correlation is significant at the 0.01 level (2-tailed). * Correlation is significant at the 0.05 level (2-tailed).

**Table 3 ijerph-21-01409-t003:** Predictors of resilience scale.

Predictors	Unstandardized B	Coefficients St. Error	Standardized Coefficients Beta	*p*-Value
Level of Depression	−1.051	0.225	−0.214	<0.001
Level of Anxiety	−0.616	0.217	−0.120	0.005
Separation	−0.810	0.551	−0.065	0.143
Marginalization	0.522	0.555	0.050	0.347
Integration	−1.129	0.443	−0.137	0.011
Assimilation	0.245	0.372	0.033	0.511
The Perception of Discrimination	0.306	0.294	0.044	0.298
Cultural Safety	2.039	0.263	0.329	<0.001
Economic Security	0.282	0.300	0.039	0.348
Physical Safety	0.871	0.299	0.121	0.004
Attitude Toward Identity	0.391	0.106	0.138	<0.001
Avoiding	0.135	0.182	0.030	0.461
Intrusion	0.185	0.166	0.044	0.265
Hyperarousal	0.250	0.214	0.047	0.243
Sociocultural Maladaptation	−0.448	0.086	−0.252	<0.001

## Data Availability

The dataset presented in this article is not publicly available due to the sensitive content. Scholarly requests to access a deidentified dataset version should be directed to the corresponding author.

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
