# Peer review of "Exploring Resilience and Its Determinants in the Forced Migration of Ukrainian Citizens: A Psychological Perspective"

_ijerph, 2024, doi:10.3390/ijerph21111409_

Round 1

Reviewer 1 Report

Comments and Suggestions for Authors

Exploring Resilience and Its Determinants in the Forced Migration of Ukrainian Citizens: A Psychological Perspective

This is a very important article for the current global situation in society.

The article enhances understanding of resilience in forced migration through a psychological lens, highlighting the importance of identifying resilience determinants and evidence-based 11 interventions.

The abstract is very well presented.

My comments are mainly related to the structure of the article, and to some content issues.

Introduction:

Please, I ask you to shorten the introduction a bit and to mark your objectives and the contribution you make.

Theoretical framework:

The article, in its current form, does not have a theoretical framework, and therefore, I ask you to use part of the introduction to make a brief theoretical framework on the psychological factors that affect refugees. There is an extensive literature on this.

Please, expand the “Stages from Trauma to Resilience” section a bit. It would be a theoretical framework together with part of the introduction.

Although I am not an expert in quantitative methodology, I consider that the methodology is very well carried out. However, as I am an expert in qualitative analysis, I would have liked to read an article with a combined methodology, since the voices of some interviewees would have given more “colour” to the article. The quantitative methodology is fine, but it does not enrich. Maybe for another work…

Likewise, I would like to highlight the quality of the article's results.

Discussion:

In this section, please, discuss only the results of your work without so many citations. The discussion seems like a theoretical framework. Please remove part of the discussion and include it in the theoretical framework mentioned above.

Conclusions:

Expand this section a little, or, if you prefer, make a link with the discussion and create a single section in which you highlight your contribution and some suggestions for future research, considering the importance of the aspect analyzed in your article.

Author Response

Our response

Reduced introduction and now mention that the theory motivated this study.

We agree that this is critical and present a quantitative result in another manuscript currently under review.

I changed the discussion and conclusion sections.

Reviewer 2 Report

Comments and Suggestions for Authors

I believe that this manuscript holds a lot of promise; it has solid documentation and a rigorous method.  My main recommendations in most of the document deal with the connection between ideas. For example, how do the ideas in the review of literature lead to the statement of purpose? How do the methods (as stated) lead to some of the findings (though for most, the connection is clear). How do the ideas in the discussion relate back to the findings from the study. In these regard, regarding some aspects, e.g., the two or three paragraphs of summary of previous lit that appears in the discussion section, the choice is ultimately up to you where the material fits. My overall impression is that at some places, the ideas presented did not drive the argument well toward the statement of purpose. My second main recommendation involves more clarity on some aspects of the method.

Beyond the main notion of connection, the writing is excellent, so I have few editorial recommendations—indeed, much fewer than I have for most manuscripts. I will break my comments into editorial/writing comments and substantive comments and questions, based on line number in the manuscript

Editorial comments:

·         51 ff: Deleting “fully,” since that treats assimilation as all or nothing, and you have already presented assimilation and separation as a dimension. For parallel structure in the list, start all defs with an “-ing” verb (maintaining, embracing, choosing, neither maintaining…) deleting extra words.

·         56: Delete “the study”

·         62: If following APA format, put sources in multiple-source citation in alphabetical order

·         68: Add “such as. . . and”: Such as Altinay and Alrawadieh (2023) and Adams and Kivlighan (2019).

·         91: “This process”—clarify: Do you mean the process of psychological support?

·         98: APA: If what follows colon is a complete sentence, capitalize

·         120: insert “that”: “…facets of resilience that displaced people may work toward”

·         345: “As the war in Ukraine continues, so does the traumatization…”

Substantive comments and questions:

·         ABSTRACT: 19-24: I cannot tell if the section “Depressive and anxiety states….and social networks” is a continued summary of findings or are more general statements about the known relationships between the concepts mentioned in the review of lit. If it is the former, make the connection clearer; if it is the latter, move it to above the statement of method and findings (or just delete it).

·         INTRO: 32ff: Clear defs of migration and nice presentation of Berry’s model with the two axes leading to four dimensions. There is no heading REVIEW OF LITERATURE, and much of this section seems actually to be the review of lit. Is “intro” the best title here (see later note on discussion of Ukraine situation, which might be a stronger intro).

·         35: Text mentions “adaptation” and “acculturation” side-by-side. Are these distinct? The same thing. I believe at some places Berry (alone and writing with Colleen Ward) distinguishes between adaptation (mental well-being) and the acculturation (navigating the norms and systems of the local culture). In much of Ward’s work, she sees two different but related aspects of adaptation. Since, as you have well-noted, there are many different defs in the literature, it is especially important to define them precisely. You do define acculturation clearly below (44-45), just as I suspect Berry would define it. Is there a source for this definition?

·         38: I appreciate the precision of noting how Berry’s approach can apply to “the acculturation experiences of immigrants and minority groups”! You then note that the first dimension is about “how individuals maintain or alter their own culture.” I’m not sure what the phrasing means here, but regarding Berry’s approach—is it about individual adaptation or about the adaptation of groups, or both? Whichever it is, make the explanation of the approach consistent throughout (see note on variables, line X, below). Consider including mention of some of the research motivated by Berry’s approach, especially if related to immigrant (or, better, “refugee”) acculturation.

·         46: “Psychological acculturation” is defined from APA dictionary, which may be appropriate. In the academic literature, Coleen Ward calls this “psychological adjustment,” and I think Berry calls it “adjustment.” Since Berry has a def, and you are using Berry, even if you stick with APA def, perhaps let reader know that you are deliberately choosing it over Berry’s term.

·         48: The text introduces Berry’s approach a second time. Consider starting with a general introduction to adaptation and its types all at once; then introduce Berry’s approach, putting the similar material in one place in the argument.

·         58ff: Consider including all you have to say about trauma before you turn to resilience. I know among psychologists, there is complex trauma and acute trauma. Consider framing the trauma of forced migration in terms of one (or both) of these items. For example, if one is leaving a land with a history of war and violence, the trauma might be “complex.” If the Ukraine has been peaceful until a sudden forced move due to an unexpected war, it might be “acute.” The def of trauma in this study is important, as some might wonder how language difficulties (line 252) would be the “most traumatic” event faced by the migrants.

·         64-65: Maybe add “and communication” (my own discipline!). Here’s a source:

Education as Catalyst for Intergenerational Refugee Family Communication About War and Trauma Lin, Nancy J.Suyemoto, Karen L.Kiang, Peter Nien-chu (2009). Resilience section might actually fit better after “positive and negative effects.” Note that the mention of the two perspectives (two types of theoretical approaches” (lines 68-70) begs the “unpacking” of these approaches (which does not appear hear) and also seems to relate more to where you introduce “theory” below (line 96).

·         72-82: This is a nice section providing the specific context in question. Maybe this could be the introduction, building a rationale for the study as a whole, after the nice opening lines you already have (moving  Berry discussion below to other  Berry discussion).

·         103 ff: Since general approaches; many modern methods; three main categories: This section shows what I find to be a tendency in the manuscript (here and in discussion section) to introduce and name perspectives or frameworks without elaboration or connection to the argument. Rather than a list of important theories that exist, I would expect a brief explanation of these and then narrowing to the theory (or method, or approach) that best relates to the present project. From my perspective, the question in a review of lit is always what the relevance is of theories, methods, research findings, etc. presented as they relate to the argument for the purpose (or research questions or hypotheses) of the current study.

·         119 ff: Three facets: Nicely detailed and clear.

·         132: “One theory proposes…” Which theory? Perhaps the theory itself is not relevant, but (if this is Victor Frankl—I don’t have the ref list with me at this moment), simply that he proposed this idea. More relevant to your argument might not be that it comes from a “theory” but that it comes from the context of Frankl, as he survived Germany’s concentration camps.

·         139: “The overall aim.” In all, while I felt that there are a lot of great ideas presented that could lead to the research purpose, they are not framed in a way that logically leads to this statement. You might be able to resolve this even with a brief paragraph that pulls the various ideas of the review of lit together to lead to the statement of purpose.

·         METHOD: 157: Here or below (probably below, after the presentation of measurement of variables), be sure to include the total length of the questionnaire and the average time it took participants to take the questionnaire (this will let readers know if response fatigue might be a concern).

·         166ff: There is nice detail on make-up and categorization of responses on the resilience scale (provide either the response options or the total range or both).

·         173: Define “stress resilience” and describe how it compares or contrasts to resilience overall. Is it a subset? This is important as it becomes a main variable in the findings, often with a variable predicting “resilience” and “stress resilience” in opposite directions. Yet a glimpse at items (e.g., resilience: “I do not dwell on things that I can’t do anything about”; stress resilience: “I can overcome any obstacles in my path”) makes it sound like the two scales could be measuring the same thing. One possibility is to include a factor analysis of all items from both scales that shows the two scales measure distinct concepts.

·         183 ff: If possible, provide more scale for this and following scales, as you have done above. For the MIRIPS, since the measure assesses “assimilation vs. separation” and “integration vs. marginalization,” clarify: Are these two measures (A/S, I/M) or four measures? If they ar measures of two dimensions (high on some aspect = assimilation, low = separation), then explain how you have assimilation and separation (and integration and marginalization) as separate variables in the findings.

·         185: “In addition, this questionnaire includes subscales to assess religion, level of religiosity, self-satisfaction, life satisfaction, and attitude toward identity.” First, I’m not sure what “attitude toward identity” means. More importantly, are these subscales of the MIRIPS or in addition. If, in addition, at a minimum, give how many items measures each and a source for the measure, though more detail would be helpful, since you do include them in your statistical analysis.

·         187: Measuring the effects of trauma: Should some discussion of the symptoms of trauma appear in the rev of lit? “The scale included three subscales to assess avoidance, intrusion, and arousal”: What are there? Since they are from the “Events Scale,” are they worded generally or to refer to a specific event?

·         In the end, there are lot of measures, which is valuable and gives great explanatory power. At the same time, some mystery remains in the manuscript about how some of the variables are conceptualized and measured, and how they relate to the argument in the review of lit. Beyond that, since there are later claims about these being high or low, it would be useful somewhere to have the number of items for each and the range of possible responses.

·         As noted above, end the data collection section with a statement of the total length of the survey (number of questions) and the average time it took participants to complete it.

·         204: Data analysis: “Predictors included cultural safety, sociocultural maladaptation…” There seems to be a mismatch between some of the items mentioned in this list and the various measures and subscales mentioned above. Just make sure nothing appears here that cannot be explicitly linked to one of the measures you have just mentioned.

·         RESULTS: 246ff: The first couple of sections of the results provide some useful and interesting information. Clarify before you move into Section 3.2 results which measures the items came from. Are these from some of the specific measures you mentioned in the methods section, or are they from single items in the questionnaire?

·         274-275: Specify the question asked or that that the numbers in the parentheses are the total number of friends reported.

·         285 ff: Here and following, many of the scores are given with notes that they are high or low. These have clear context in terms of resilience, as the methods section indicates a “high” score for that variable. Without context of the total ranges, the means provided here and in some of the tables are difficult for the reader to interpret.

·         290: “Which is higher than than in the general population of adults: Source?

·         294: Acculturation strategies might have a new paragraph. As per my note above, I would avoid the word “fully,” since this seems to be a continuum. Maybe talk not just about acculturation in this paragraph, but the relative scores on the two dimensions (or four separate factors, depending on how you measured A/S and I/M—see note at 183).

·         Table 1: Consider grouping the variable with subheadings, left-flush margin for first column for readability; make sure no variables appear in this list that are not explicitly mentioned in the methods section (overall trauma rate? Cultural safety? Depression? Anxiety?)

·         302 ff: The findings are quite interesting. Consider moving the explanation of the findings (302 ff) to the discussion section. The identity struggle over time, in which longer time in the country might “erode resilience” is interesting. The findings remind me of an older study (Tanaka et al., 1994) in Japan in which the longer the sojourner was in Japan (up to 3 years) and the better their Japanese, the more isolated they felt, as they realized that they were outsiders.

·          315: “a direct negative effect on…” I am not a stats expert, so I really don’t know the answer to this question: Do the statistics used in the study allow for a causal interpretation, or are they correlational statistics? Here and in next paragraph, save explanations of findings for discussion. The discussion of the role of environment reminds me of Young Yun Kim’s (2005 and other years) theory of adaptation which includes the role of “host receptivity” and an article by Sam and Berry that talks about the impact of the receiving culture’s view of immigrants (or ethnic groups) and the acculturative choices of the individuals in those groups.

·         325-336: Much of this seems to belong in the discussion.

·         339: It’s good to start by reminding the reader of the situational context that drives the study. Beyond this, much of the more general discussion of previous literature and concepts from 348-388 sound more like “review of lit” concepts, as long as the ideas are relevant to the argument (see note at 103ff). I often expect the discussion (after a brief intro to the section, like you already have) to move into a summary and theorization of the findings.

·         396: “consistent with other research…” Which? (cite)

·         401: “supportive conditions.” Consider Berry & Sam (2010), Kim’s (2005) idea of host receptivity.

·         411: “surviving the trauma of war” raises a whole new question, but one that is important: Those fleeing The Ukraine have already experienced the “trauma” of war. Is that, or the “traumas” of living in a new culture what is at stake here?

·         416 ff: The text mentions several individual characteristics related to resilience. How do these relate to the findings in the present study? (e.g., are they the same as your variables? Do they point to future research?)

·         The remainder of the implications and limitation are well-written. Consider the potential limitation of response fatigue, depending on the length of the survey.

Author Response

Thank you for your review

Round 2

Reviewer 2 Report

Comments and Suggestions for Authors

I feel that the manuscript is much improved in its overall cohesion and clarity. I have comments below, but many are recommendations. I feel that you have done a diligent job making changes, including some major moving around and deletions to create a tighter, more cohesive manuscript. As always, my suggestions are below:

EDITING/WRITING

·         44: Remove hyphen after the period.

·         46: After colon, add some phrasing that admits that these are only examples of the theories—something like: These include…” Add an “and” before the last item in the list.

·         47: Add “the” before “ecological…”

·         119: “To contextualize these ideas within trauma theory [24] identifies six broad theoretical frameworks”—the subject of the main clause is missing (who identifies…)

·         127: Possibly start a new paragraph as you move to PTSD. I think that would make it stand our more as the approach that most guides your study (from trauma).

·         179: 3 studies. I think APA would recommend these be in alphabetical order by first author of each source—at least that’s the rule if they’re inside parentheses.

·         Ironically, I was dismayed at seeing a number of typos in *my own review* from the first submission  ☹

SUBSTANTIVE COMMENTS

REV OF LIT

·         45: I still think some subheadings would be useful to guide the reader through the text, if not “review of literature” with subheadings, then some other subheadings.

·         53: The phrase “happening when individuals or groups choose to” seems to apply not just to this strategy, but to all four of the strategies. So move this out of the third item and state *before* any of the four strategies that they might apply to either individuals or groups. Delete “(referring to a situation where individuals” from the last item. Start the parenthetical definition with “retaining”  and the last parenthetical def with “(neither maintaining nor…). This will both clarify the description of the strategies and create parallelism in the grammar for the definitions.

·         79ff: Deleting the text perhaps saves you from having to define and defend the definition of psychological adaptation here. My main recommendation here is to clearly define trauma (in general) before you move to discuss “trauma as the result of forced migration.” That might also help the readers understand the ways in which such trauma might be characterized by either “acute” or “complex” trauma (note: I am typing this as I read through the paper, so please keep reading before you change anything!).  While these divisions make sense to me, I do see below that you clearly frame the trauma here as possibly relating to PTSD, which seems an appropriate approach. Oh… I just got to the discussion of types of trauma related to war refugees. Yes! This section is quite nice and addresses one of my suggestions from the first draft. I think that outlining the types of trauma where you do is sufficient and makes sense.

·         111: Ah—here is a new section that lays out strengths and limitations of trauma. This is good. It still seems to fit before you narrow to “trauma as the result of forced migration” (a “broad-to-narrow” organization of ideas) and seems to imply that you have first defined what, exactly, trauma is (something I don’t see yet in the paper, but that seems central to what you are doing here).

·         119ff: Good. You still list six approaches to trauma, but you clearly note the one that you will use and why. This addresses one of my suggestions from the first draft (listing of theories with no link to the present study).

·         234: Nice revision of discussion of Frankl and his writing context.

·         Overall, aside from some notes above, the Rev of Lit flows well and is well-documented in previous literature.

METHOD:

·         302: There is now a clear def and discussion of “stress resilience.”

·         319: MS is much clearer on how you handled measuring Berry’s four acculturative strategies. To be honest, I don’t know how Berry and his colleagues measured it; what matters here is that you have explained clearly what *you* did. (If this reflects how Berry and others do it, all the better! If not, *consider* defending your measure of each strategy separately, rather than as two dimensions).

·         335: My main suggestion here is to give more detail on the subscales of the MIRPS and how the measured variables relate to the theorization you have above. The items here seem to come up as surprises.

·         349ff: I like the text you added to clarify what people are responding to on the PTDS measure. And I see the inclusion of the total number of items and time to take the survey. I doubt that response fatigue would be an issue if the survey is only 30 minutes.

RESULTS

·         It looks like the revision addresses most or all of my suggestions on the first draft (e.g., including ranges to contextualize statements of a “high” or “low” score, speaking in terms of association if the statistic does not allow determination of causality; movement of “explanation” of findings to discussion section). I have no new recommendations, here.

DISCUSSION:  

·         I think overall the discussion is much more balanced without extensive “review of lit” that might go before the method and an increase of explanation of findings.

·         697: Consider adding a phrase here, based on the context of your study, that makes it clear that you are applying the findings and implications specifically to those who experience trauma based on forced migration (e.g., in war situations), though your findings may have relevance beyond to other situations of high trauma and need for resilience.

Author Response

Thank you for your valuable feedback on the manuscript. I appreciate your attention to detail and have made the following revisions in response to your suggestions:

  • Line 44: The hyphen after the period has been removed.

  • Line 46: I have added phrasing to clarify that the theories mentioned are examples, placed an “and” before the last item in the list.

  • Line 47: The article “the” has been added before “ecological.”

  • Line 119: I clarified the subject of the main clause by specifying that the authors identify the six broad theoretical frameworks.

  • Line 127: I have started a new paragraph to better delineate the discussion on PTSD, highlighting its significance as the primary approach guiding the study.

  • Line 179: The three studies have been reordered alphabetically by the first author in accordance with APA guidelines.

Overall Comments:

  • I considered incorporating subheadings to enhance the organization of the text and guide the reader through the literature review. This change aimed to clarify the structure and key points.

Specific Edits:

  • Line 53: I revised the introduction to the four strategies to indicate that they applied to both individuals and groups. I removed the phrase "referring to a situation where individuals" from the last item and started the parenthetical definitions with "retaining" and "neither maintaining nor..." to enhance clarity and parallelism.

  • Lines 79ff: I agreed that defining trauma in general before discussing “trauma as the result of forced migration” would provide important context. I ensured that a clear definition was included and differentiated between “acute” and “complex” trauma. I appreciated your positive feedback regarding the section on types of trauma related to war refugees; I was glad it addressed your earlier suggestions.

  • Line 111: I revised the organization of the manuscript to define what trauma was before narrowing the focus to “trauma as the result of forced migration.” This change created a more coherent flow of ideas.

  • Lines 119ff: I appreciated your recognition of the clarity regarding the six approaches to trauma and the connection to my study. I maintained this structure while ensuring the rationale for the chosen approach was evident.

  • Line 234: Thank you for your positive comment on the revision of the discussion of Frankl and his writing context.

  • Method Section:

    • Line 302: I included a clear definition and discussion of “stress resilience.”

    • Line 319: I clarified how I measured Berry’s four acculturative strategies and, if necessary, provided additional defense for my measurements.

    • Line 335: I provided more detail on the subscales of the MIRPS and clarified how the measured variables related to the theoretical framework outlined earlier in the paper.

    • Lines 349ff: I was glad the additions clarifying what respondents were answering in the PTSD measure were helpful. I ensured that the information regarding the survey duration was communicated effectively.

Results:

  • Thank you for acknowledging that the revisions addressed your previous suggestions regarding the results section. I appreciated your positive feedback.

Discussion:

  • I was pleased to hear that you found the discussion more balanced and focused on the findings rather than an extensive review of literature.

  • Line 697: I added a phrase to specify that the findings and implications were particularly relevant to individuals experiencing trauma from forced migration in war situations, while also noting the broader relevance of these findings to other high-trauma contexts.

Thank you again for your thorough review. I believe these changes have improved the clarity and flow of the manuscript.